# Plant Hormone Modularity and the Survival-Reproduction Trade-Off

**DOI:** 10.3390/biology12081143

**Published:** 2023-08-17

**Authors:** Jasmina Kurepa, Jan A. Smalle

**Affiliations:** Plant Physiology, Biochemistry, Molecular Biology Program, Department of Plant and Soil Sciences, University of Kentucky, Lexington, KY 40546, USA; jasmina.kurepa@uky.edu

**Keywords:** biological modules, survival-reproduction trade-off, reproduction maximization, reproduction assurance, cytokinin, plant hormones, double-negative signaling

## Abstract

**Simple Summary:**

In this review, we explore the modularity of plant hormones and their role in the trade-off between survival and reproduction. Biological modularity refers to the organization of living organisms into distinct units that work together to ensure individual and species’ well-being. Our findings reveal that plant hormones can be classified into different modules, each with its specific function in the reproduction trade-off. The cytokinin module, for example, focuses on maximizing reproduction, while other hormone modules function to assure reproduction. Through this review, we present examples of the modular nature of plant hormones and highlight their crucial role in balancing survival and reproduction in plants.

**Abstract:**

Biological modularity refers to the organization of living systems into separate functional units that interact in different combinations to promote individual well-being and species survival. Modularity provides a framework for generating and selecting variations that can lead to adaptive evolution. While the exact mechanisms underlying the evolution of modularity are still being explored, it is believed that the pressure of conflicting demands on limited resources is a primary selection force. One prominent example of conflicting demands is the trade-off between survival and reproduction. In this review, we explore the available evidence regarding the modularity of plant hormones within the context of the survival-reproduction trade-off. Our findings reveal that the cytokinin module is dedicated to maximizing reproduction, while the remaining hormone modules function to ensure reproduction. The signaling mechanisms of these hormone modules reflect their roles in this survival-reproduction trade-off. While the cytokinin response pathway exhibits a sequence of activation events that aligns with the developmental robustness expected from a hormone focused on reproduction, the remaining hormone modules employ double-negative signaling mechanisms, which reflects the necessity to prevent the excessive allocation of resources to survival.

## 1. Introduction

Trade-offs, inherent in all living organisms, involve the delicate balance between opposing yet crucial processes [1]. One fundamental trade-off centers around the survival of an individual organism versus the survival of the species, which relies on successful reproduction [2]. Reproduction utilizes resources that could otherwise be allocated for optimal adaptation to the environment, posing a threat to individual survival. Conversely, excessive resource allocation towards well-being and survival limits reproduction, which is disadvantageous in the competitive realm of species. Extensive research across various species has unveiled sophisticated mechanisms that establish the equilibrium between reproduction and survival in the natural world, favoring resource allocation towards reproduction [3,4,5,6,7]. These studies consistently indicate that resources are dedicated to individual survival only to the extent that reproduction can be maximized and fulfilled. In species other than plants, hormonal regulation was shown to be crucial in maintaining a balance between reproduction and survival [7,8,9,10,11]. These hormones govern physiological and developmental processes related to either reproduction or survival, ensuring an equilibrium between the two.

Biological modules are discrete components within a biological system that perform specific functions, often interconnected and working together to coordinate organismal functions [12,13]. These modules exist at various levels of organization, spanning from molecular and cellular modules to organ-level modules. Modularity typically arises when two groups of traits are selected in opposite directions, such as in the context of the survival-reproduction trade-off [12,13]. While trade-offs drive the emergence of modularity in biological systems, modularity is also associated with traits that are selected in a similar direction, regulating distinct aspects of the same selection goal, for example ensuring survival by maintaining tolerance to a wide range of different stress conditions that require different sets of solutions. Modularity enhances the adaptive capabilities of biological systems by allowing targeted and independent evolutionary changes within different modules [13].

In this review, we outline the hormonal modularity responsible for governing the delicate balance between survival and reproduction in plants. Our findings reveal that among the various hormone modules, cytokinin (CK) uniquely takes center stage in maximizing reproduction. This distinctive role of the CK module is evident through its direct response activation signaling pathway, setting it apart from all other hormone modules, such as auxin, brassinosteroids, gibberellic acid, ethylene, jasmonic acid, strigolactone, abscisic acid, and salicylic acid, which encompass a double-negative signaling mechanism. We explore how these two fundamentally different signaling mechanisms align with the regulation of the survival-reproduction trade-off by plant hormone modules. Finally, we put forward a proposed pathway for hormone modular emergence, identifying the CK/auxin pair as the primordial drivers for how plants navigate the intricate balancing act between well-being and reproduction.

## 2. The Survival-Reproduction Trade-Off in Plants

Before delving into the role of each hormone module in the trade-off between survival and reproduction, it is important to address certain processes and concepts that differentiate the survival-reproduction trade-off in plants from that of other organisms. Notably, plants possess unique adaptations that enable them to sustain growth and reproduction even in the face of heightened stress conditions. Unlike mobile organisms, plants are unable to escape unfavorable conditions, thus relying on growth and metabolic adjustments to cope with their environment. In contrast to most non-plant species, which respond to stress by entering a quiescent state, plants ensure their survival by modifying growth and, in many cases, sustaining growth at a reduced rate [14]. As a result, when discussing plants, it is more accurate to use the terms “reproduction assurance” under elevated stress conditions and “reproduction maximization” under optimal conditions instead of the conventional “survival-reproduction trade-off”. These more precise descriptors better capture the dynamics of plant responses to varying environmental circumstances, highlighting the strategic choices plants make concerning reproduction in different contexts. This reproduction assurance versus reproduction maximization balance will be referred to as the reproduction trade-off throughout the text.

### 2.1. Direct Reproduction Assurance, Indirect Reproduction Assurance, and Reproduction Maximization—Definitions

We can make a distinction between direct and indirect reproduction assurance strategies (Figure 1). Direct strategies involve metabolic and growth adjustments that enable reproduction under elevated stress conditions, albeit with reduced reproductive output. On the other hand, indirect reproduction assurance strategies encompass growth and physiological changes that delay reproduction until environmental conditions become more favorable. Depending on specific environmental cues, indirect strategies may eventually give way to direct strategies, ensuring successful reproduction under prolonged environmental stress.

Hence, it is crucial to perceive the connection between survival and reproduction in plants as a continuum rather than a binary distinction. It represents a spectrum wherein one end emphasizes reproduction maximization, the other end focuses on indirect reproduction assurance, and different forms of direct reproduction assurance occupy the space in between. This understanding implies a diverse range of strategies and resource allocations between survival and reproduction, with varying degrees of emphasis depending on the prevailing environmental conditions.

### 2.2. Most Prominent Processes Involved in the Reproduction Assurance-Reproduction Maximization Trade-Off

Based on the prevailing environmental conditions, plants employ a diverse array of developmental and physiological strategies to effectively balance reproduction assurance and reproduction maximization. Some key examples are given in the following subsection.

#### 2.2.1. Co-Regulation of Photosynthesis and Reproduction

Considering that photosynthesis is the main supplier of energy and resources for the development of reproductive structures in plants, it is expected that higher rates of photosynthesis would positively impact reproductive success. The connection between photosynthesis, reproduction, and environmental conditions dates back to the early stages of the evolutionary history of plants. For instance, in unicellular photosynthetic algae, elevated photosynthetic rates are positively associated with rates of binary fission (i.e., cell division), while they are negatively correlated with nutrient availability [15]. Studies on mutants of *Chlamydomonas reinhardtii* that are unable to suppress photosynthesis in response to nutrient deprivation reveal an increased proliferation rate under optimal nutrient conditions but a reduced survival rate under phosphate-deprived conditions [16].

#### 2.2.2. Regulation of the Shoot-to-Root Growth Ratio

Since the shoot is the main photosynthetic organ and the main organ responsible for both sexual and asexual reproduction, it is expected that an increased shoot-to-root growth ratio translates into increased reproduction potential, as this represents a maximized carbon allocation for reproductive development and sustenance [17,18]. Therefore, control of the plant’s shoot-to-root growth ratio is an important reproduction trade-off. The regulation of the shoot-to-root growth ratio is an evolutionary response to the greater water and nutrient stresses faced by land plants compared to their algal ancestors [18]. In response to these stresses, plants reallocate resources by inhibiting shoot growth, thereby delaying or minimizing reproduction, in favor of promoting root growth. This reallocation of resources ensures improved hydration and nutritional well-being and also reproductive success in the post-stress period. Inhibition of shoot growth is also an unintended consequence of resource allocation towards enhancing tolerance to abiotic and biotic stresses, and this is generally referred to as the growth-defense trade-off [19,20,21].

#### 2.2.3. Timing of Senescence

Senescence is a process primarily driven by nutrient status and it ensures the reallocation of nutrients based on developmental and environmentally imposed needs [22]. At the core of senescence are the controlled degradation of the photosynthetic apparatus, the main nutrient-containing part of leaf cells, and the loading of its nutrient constituents into the phloem for transport toward beneficiary organs [23]. Therefore, senescence acts as a mechanism that inhibits photosynthesis and shoot growth, prioritizing resource allocation toward reproductive development. This process plays a critical role in direct reproduction assurance by enabling the reproductive development of plants facing elevated stress levels that pose a threat to shoot and reproductive well-being [23].

#### 2.2.4. Regulation of Flowering Time

Flowering time is a critical trait that influences the survival and reproductive success of plant species. Flowering, the transition from vegetative to reproductive growth, limits the size of the shoot and limits the resources that can be allocated to maximal seed yield. However, flowering is obviously also needed for reproduction, and actions that delay flowering will negatively impact the length of the reproduction phase. Thus, flowering time can be viewed as an important reproduction trade-off. From the perspective of maximizing reproduction, early flowering will be beneficial only if it is accompanied by extending the length of the reproduction phase. There are two distinct forms of accelerated flowering that warrant examination in this context (Figure 2). The first type entails accelerated flowering coupled with delayed senescence, which extends the reproductive phase and signifies a focus on reproduction maximization. Conversely, the second type encompasses accelerated flowering without senescence inhibition or in combination with senescence acceleration, representing direct strategies for reproduction assurance in response to environmental stresses.

The regulation of the timing of entry into the reproductive phase in response to stress is an ancient evolutionary trait. For example, in algae, both sexual and asexual reproduction can be triggered by wounding or other forms of stress [24].

#### 2.2.5. Dormancy

This aspect of plant developmental programming includes shoot growth arrest and the suppression of seed germination [25,26,27]. Dormancy can be triggered by elevated stress levels, but it also plays a crucial role in the anticipation of impending lethal stress, such as winter, through circadian clock control and photoperiodic cues. Consequently, dormancy can be categorized as a component of indirect reproduction assurance, as it ensures plant survival, thereby preserving the potential for future reproduction.

#### 2.2.6. Determination of Seed Size

Seed size exhibits significant variation among angiosperms and these variations in seed size exemplify another aspect of the reproduction trade-off. Smaller seeds prioritize maximal reproduction potential relative to resource allocation [28]. On the other hand, larger seeds serve as a form of indirect reproduction assurance by providing seedlings with ample nutrients during early development [28]. This becomes particularly crucial when germination takes place in nutrient-deficient conditions, leading to higher survival rates and an enhanced potential for reproduction [28]. The size of the endosperm assumes a crucial role in this context, as it determines the nutrient status of the seed and ultimately affects the viability of developing and germinating seedlings under conditions of nutrient deficiency [29,30].

## 3. Cytokinins (CKs) and Reproduction Maximization

Based on the strategies that influence the balance between survival and reproduction, we can propose that a hormone module responsible for reproduction maximization would enhance photosynthesis, promote shoot growth, and increase seed yield while inhibiting root growth, senescence, and limiting seed size. Additionally, the reproduction maximization hormone module is expected to confer heightened sensitivity to drought and nutrient deficiency stresses and be subject to quantitative negative control mechanisms under these stress conditions.

When examining the functions of CK and other plant hormones within the context of this trade-off, it is crucial to prioritize research findings that are anticipated to provide the most accurate representation of hormone modular function. Taking this into consideration, we have given greater significance to research findings based on genetic loss-of-function approaches rather than gain-of-function studies. The latter often involves the use of transgenes with strong promoters that can lead to non-native expression of investigated genes, which may not accurately reflect the actual function of a specific module. Similarly, we have cautiously approached hormone treatment studies, as they involve artificially manipulated hormone accumulation and distribution levels that may not align with natural hormone regulation within the plant. Another factor to consider is the extent to which loss-of-function mutations affect hormone sensitivity or biosynthesis. Severe reductions in sensitivity or biosynthesis may provide limited insights compared to a partial loss of function, especially in hormones involved in plant growth promotion, such as CK, auxin, GA, and BRs. Severe loss of function in these hormones can result in significant dwarfism, making it challenging to determine their specific roles in growth regulation [31,32,33,34]. Furthermore, it is important to differentiate between involvement in reproductive development and overall reproduction maximization. Just because a hormone is necessary for reproductive development does not necessarily mean it plays a direct role in maximizing reproduction.

With these caveats in mind, and upon examining the known functions of different plant hormones, it becomes evident that CKs play a prominent role as the hormone module for reproduction maximization. In addition to its function in processes crucial for reproduction maximization, CKs also promote resource allocation for direct reproduction assurance. In light of this, we will explore the specific functions of CKs in relation to reproduction maximization and reproduction assurance separately.

### 3.1. Reproduction Maximization

The importance of CKs in higher plant reproduction was initially established through genetic studies conducted on *Arabidopsis thaliana*. These studies have provided evidence indicating that CKs promote shoot growth, inhibit root growth, and delay senescence, which in combination with the acceleration of flowering time, maximizes reproduction [35,36,37]. Importantly, the positive impact of CKs on sexual reproduction is not limited to dicotyledonous plants like Arabidopsis. Similar effects have been observed in monocotyledonous plants such as rice, as well as in primitive plants like *Marchantia polymorpha* and *Physcomitrella patens* [38]. These findings highlight the conserved role of CKs in maximizing reproductive success across diverse plant species.

CKs are also promotors of all known forms of plant asexual reproduction. For example, stolons and rhizomes are shoot organs that initiate and grow either below or in close contact with the soil surface, and they are responsible for the formation of new plants that derive from the originating mother plant that forms these shoot structures [39]. CKs induce the initiation of stolons and rhizomes, and CKs play an essential promotion role in downstream processes that involve tuber initiation from stolons and shoot development from tubers and rhizomal nodes [40,41,42,43,44].

CKs also play a significant role in the asexual reproduction of duckweed. Duckweeds, which are aquatic plants, exhibit both sexual and asexual reproduction, with asexual reproduction being more prominent and achieved through frond formation [45]. In duckweeds, a frond serves as the functional equivalent of leaves in terrestrial plants. It is a disk-shaped structure that carries out photosynthesis and possesses air pockets that enable it to float on the water’s surface. Asexual reproduction in duckweeds occurs through the development of new fronds from meristematic regions on the mother frond. These newly formed fronds eventually detach from the mother frond, giving rise to clonal and independent duckweed plants. This process plays a crucial role in the rapid coverage of water surfaces by duckweeds. CKs exert a strong promoting effect on frond formation and substantially accelerate this type of vegetative propagation [46].

The reproductive role of the CK module has been observed in unicellular algae as well. In these organisms, CKs promote photosynthesis and vegetative propagation through cellular duplication (binary fission), indicating that their evolutionary origins are ancient and closely tied to reproduction in photosynthesizing organisms [38]. While the initial discovery of CKs as cell division promoters already hinted at their connection to reproduction, the extensive evidence demonstrating their positive regulation of various aspects of sexual and asexual reproduction in all photosynthesizing organisms suggests that CKs can be regarded as a broad-spectrum hormone module focused on maximizing reproduction.

Indirect indications of this reproductive focus can be observed in the detrimental impact of CK on a plant’s tolerance to drought and nutrient deficiency stress [18]. This suggests that plant survival is not the primary focus of CKs and that the CK module requires down-regulation in response to these stresses.

### 3.2. Direct Reproduction Assurances

While CKs primarily exhibit a reproduction maximization role compared to other plant hormone modules, they are also implicated in reproduction assurance. However, their involvement is limited to the realm of direct reproductive assurance. Two categories of direct reproduction assurance involving CK can be distinguished, and both provide additional indirect evidence for the reproductive focus of CK. The first category involves CK’s promotion of a stress tolerance mechanism while also imposing limitations to ensure optimal carbon allocation for shoot growth. In such cases, high levels of stress often result in the repression of the CK module, leading to increased stress tolerance and a decrease in the shoot-to-root growth ratio, which is attributed to reduced CK action. In the second category, CK promotes high tolerance levels to stresses that cannot be alleviated by adjusting the shoot-to-root growth ratio.

An example belonging to the first category involves the interplay between CK and anthocyanin biosynthesis. Anthocyanins function as antioxidants, safeguarding against damage caused by excessive reactive oxygen species (ROS) produced during photosynthesis [17]. CK promotes anthocyanin biosynthesis, albeit to a moderate extent, and also limits anthocyanin biosynthesis to what is needed for shoot growth under optimal conditions [17]. This moderation in anthocyanin biosynthesis under normal growth conditions is crucial because this metabolic pathway requires a significant amount of carbon resources, which would otherwise be diverted from shoot growth [17]. The limiting effect of CK on this pathway becomes evident when plants experience elevated levels of nutrient deficiency and drought stress. These conditions induce heightened anthocyanin biosynthesis as a response to the increased oxidative stress associated with these stresses. These conditions also downregulate CK action, thus counteracting the CK-dependent inhibition of anthocyanin synthesis and allowing for the accumulation of high levels of anthocyanin while also promoting a reduction in the shoot-to-root growth ratio [17,18].

The role of CKs in symbiotic interactions with arbuscular mycorrhizal fungi and Rhizobium bacteria exemplifies another instance of direct reproduction assurance. These interactions enhance the plants’ access to phosphorus and the legumes’ access to nitrogen, respectively, leading to improved reproductive outcomes [17]. CKs play a positive role in establishing these interactions, but it also restricts them when nutrient homeostasis is optimal to support shoot growth. By doing so, CKs again maximize the allocation of carbon resources for reproduction [17]. Under high levels of nutrient deficiency, CK action is reduced and the limitation imposed by CK on symbiotic interactions is diminished. This results in increased root growth and enhanced symbiosis, accompanied by decreased shoot growth, ultimately ensuring the survival of the plant.

The second category of reproduction assurance is best illustrated by CK’s contribution to heat shock and heat stress tolerance, and plant immunity [47,48,49]. Again, these actions can be interpreted as reproduction focused as they involve resource allocation that promotes reproduction maximization under stress conditions. Heat shock stress poses a significant risk to plant survival, and unlike drought and nutrient deficiency stresses, reducing the shoot-to-root growth ratio does not relieve the threat to the plant in this case. Similarly, pathogen infection poses a lethal threat, and altering the shoot-to-root growth ratio does not offer a solution (Figure 3).

## 4. Hormones with Reduced Reproduction Trade-Off Ratios

In contrast to CKs, which can be seen as the main reproduction maximization module, the other hormone modules exhibit a decreasing reproduction trade-off ratio. Among these, BRs function as a stress-induced and weaker-than-CK version of the reproduction maximization module. The other classical hormones do not actively participate in reproduction maximization. Instead, they regulate diverse mechanisms that promote direct and indirect reproduction assurance. In this section, we present a succinct overview of hormone actions closely associated with reproduction. To delve deeper into the intricate mechanisms through which these hormones govern plant development and physiology, we direct readers to the numerous comprehensive reviews that are readily accessible. These reviews, due to their abundance, are not individually listed.

### 4.1. Brassinosteroids (BRs)

#### 4.1.1. Reproduction Maximization

Like CKs, BRs play positive regulatory roles in photosynthesis, shoot growth, and reproduction [50,51]. However, contrary to CK, BRs delay flowering time, thus limiting the reproductive phase [52]. Interestingly, the effects of BR and CK on seed yield appear to be synergistic as their combined activities result in a further increase in seed number [53]. BRs also promote photosynthesis and asexual reproduction in duckweeds [54]. Consequently, BRs can be considered as a legitimate reproduction maximization module, albeit to a lesser extent than CK.

#### 4.1.2. Direct Reproduction Assurance

Compared to CKs, the direct reproduction assurance functions of BRs are more pronounced, thus diverting more resources away from reproduction maximization. Firstly, in contrast to CKs, BRs promote root growth, which diverts resources away from shoot development and maximal reproductive growth [55]. Secondly, and also in contrast to CKs, BRs promote senescence, which limits shoot photosynthetic capacity [56]. Finally, and again the opposite of CKs, BRs enhance endosperm growth and, thus, seed size, resulting in greater resource consumption per genome copy [30,57]. These three effects, despite their negative impact on reproduction, are intricately linked to the crucial role of BRs in enhancing tolerance to various environmental stresses. Two types of stresses underscore a fundamental difference between the functions of BRs and CKs [51]. While both BRs and CKs contribute to heat and biotic stress tolerance, BRs also play a role in promoting tolerance to drought and nutrient deficiency stresses, where CKs have negative effects [18,55]. This is evident from the activation of BR biosynthesis and signaling pathways under these stress conditions, while CK activity is suppressed. Furthermore, the promotion of seed size by BRs can be seen as a strategy for stress tolerance, as larger seeds provide more resources for early seedling growth, especially under nutrient-deficient conditions [28]. Hence, it can be inferred that BRs represent a stress-responsive reproduction maximization module, prioritizing reproduction under drought and nutrient deficiency conditions that require the repression of the CK module. Interestingly, this stress-induced reproduction focus of BRs is specific to land plants, which face higher levels of drought and nutrient deficiency stresses compared to algae, where CK regulation is also present [38,58]. Figure 4 illustrates a comparison between CK and BR in promoting reproduction relative to drought and nutrient deficiencies.

While BRs promote tolerance to drought stress, their activities in promoting shoot growth and reproduction can pose risks to the plant under high levels of drought stress, as the shoot is the main water-consuming part. As a result, under conditions of high drought stress, the action of BR is suppressed, providing further evidence that this hormone is primarily focused on reproduction and should be inhibited when the plant’s survival is in jeopardy [52].

### 4.2. Auxins

#### 4.2.1. Direct Reproduction Assurance

Auxins play a vital role in mediating differential growth responses, enabling plants to adapt their growth patterns in response to gravity, directional light, and other stimuli [59]. Of particular interest is the role of auxins in the shade avoidance growth response. In this response, auxins stimulate an increase in the shoot-to-root growth ratio, which is in contrast to their function in plants grown under optimal light conditions [60]. It is important to recognize shade avoidance as a stress response, as plants strive to overcome a temporary suboptimal light environment by promoting the elongation of stems and leaves [60]. Consequently, resources are redirected towards this process, prioritizing vegetative shoot growth rather than maximizing reproduction, especially since the elongated parts of the shoot (e.g., stems and leaf petioles) contribute less to photosynthesis than leaves. Nevertheless, facilitating the shoot’s access to a more favorable light environment can be considered a form of direct reproduction assurance, as the shoot is essential for reproductive growth.

Another form of direct reproduction assurance by auxin involves its stimulatory action on endosperm growth, wherein it acts antagonistically to CK, thus potentially ensuring the survival chances of germinating seedlings subjected to nutrient deficiency stress [30].

#### 4.2.2. Indirect Reproduction Assurance

One of the main roles of auxins is directly related to their negative impact on reproduction maximization. Auxins play a crucial role in supporting plant well-being, particularly under challenging conditions such as drought and nutrient deficiency [18]. Its primary function in these circumstances is to promote root growth and help establish symbiotic interactions with arbuscular mycorrhizal fungi (for phosphorus uptake) and with Rhizobium bacteria (for nitrogen fixation in legumes) [18]. However, a trade-off is observed as auxins have the simultaneous effect of inhibiting shoot growth and suppressing the biosynthesis and activity of CKs [17,18]. In the context of auxin’s impact on senescence, it is important to note that auxin effects primarily target shoot viability, indicating its role in indirect reproduction assurance. This delay in senescence is accompanied by a reduction in overall reproductive growth [61]. This prioritization of resource allocation towards root growth and symbiotic relationships over shoot growth is essential for plant survival under unfavorable conditions.

### 4.3. Abscisic Acid (ABA)

#### 4.3.1. Direct Reproduction Assurance

The role of ABA in ensuring plant well-being directly impacts reproduction maximization. During drought, as well as some other stresses, it plays a vital role in promoting stomatal closure [62], which leads to an overall reduction in photosynthesis. Furthermore, ABA is involved in the induction of flowering under long-day conditions in response to drought stress [63], which, combined with its role in promoting senescence [56], further restricts reproduction.

#### 4.3.2. Indirect Reproduction Assurance

ABA plays a significant role in enhancing long-term reproductive success by inhibiting flowering in response to drought stress during short days [63]. Additionally, it ensures the viability of reproduction over extended periods by inducing dormancy in shoot buds and seeds [64,65]. This dormancy prevents shoot growth and seed germination under harsh conditions like severe winter and extreme drought, thus protecting the survival and reproductive potential of the plant.

### 4.4. Gibberellic Acid (GA) and Direct Reproduction Assurance

GAs appeared simultaneously with the development of vascular systems in land plants, which allowed the upwards growth of a shoot [66,67]. While GA promotes both shoot growth and photosynthesis [67,68], it cannot be classified as a hormone focused on reproduction. The stimulatory effect of GA on shoot growth is primarily limited to vegetative growth rather than reproductive growth, as it mainly influences the elongation of stems and leaf petioles. Consequently, GA can be more accurately categorized as a module for direct reproduction assurance due to its primary role in preserving or enhancing an optimal light environment for shoot growth. A crucial aspect of GA’s function is its involvement in facilitating the shade-avoidance response in plants, achieved by promoting the degradation of DELLA proteins that inhibit the shade-avoidance response [69,70]. By engaging in shade-avoidance growth, which combines accelerated stem elongation and the suppression of shoot branching, plants can effectively compete for optimal light conditions. It is not surprising that GA biosynthesis is upregulated in response to shade, particularly in the presence of an increased far-red to red light ratio [71]

Further evidence supporting the negative impact of GA on reproduction maximization can be found in recent instances of plant domestication linked to the green revolution. The development and utilization of semi-dwarf grain varieties with reduced GA action have resulted in an increased grain-to-straw biomass ratio, confirming the resource diversion from reproduction to vegetative shoot growth caused by GA [72].

### 4.5. Ethylene and Direct Reproduction Assurance

Some of the well-being actions of ethylene are directly related to how it limits reproduction maximization. Ethylene acts as a general growth inhibitor, accelerates flowering time, and promotes shoot senescence, thereby reducing photosynthesis and reproductive potential [73]. However, these actions contribute to direct reproduction assurance in response to a range of stresses that jeopardize plant and reproductive viability.

### 4.6. Strigolactones (SLs) and Reproduction Assurance

SLs restrict the maximization of reproduction by inhibiting shoot branching, promoting shoot senescence, and further suppressing photosynthesis through the inhibition of stomatal formation and conductance [74,75,76]. Mutants with impaired SL biosynthesis exhibit increased stomatal density, higher photosynthesis rates, and enhanced seed yield. The limitations imposed by SL on reproduction are attributed to its role in both direct and indirect reproduction assurance, which are intricately linked and challenging to separate. SL contributes to salt, osmotic, and drought stress tolerance, particularly in the shoot, partly due to its negative influence on stomatal formation and conductance [77]. In addition, SLs play a critical role in helping plants cope with nutrient deficiency stresses by working in conjunction with auxin to regulate the shoot-to-root growth ratio, control root development, and facilitate beneficial interactions with Rhizobium bacteria and arbuscular mycorrhizal fungi [74]. Furthermore, SLs accelerate leaf senescence in response to nutrient deficiency stresses, further suppressing shoot growth in favor of root growth [74]. Lastly, SLs participate in the shade-avoidance response by inhibiting shoot branching when exposed to high far red-to-red light ratios [78].

### 4.7. Jasmonate (JA)

#### 4.7.1. Direct reproduction assurance

JA plays crucial roles in safeguarding the shoot against oxidative stress, as well as in enhancing plant resistance to pathogens and in the response to herbivory and wounding stress [79,80].

#### 4.7.2. Indirect Reproduction Assurance

JA negatively affects reproduction both indirectly by suppressing shoot growth and promoting leaf senescence and directly by directly repressing reproductive development and delaying flowering time [81,82,83]. Furthermore, the anti-reproductive effects of JA are evident in its ability to inhibit cell division [84,85]. The negative effects of JA on reproduction are closely linked to one of its primary functions, which is to restrict shoot and reproductive development in response to drought stress, a known trigger of JA biosynthesis [86,87].

### 4.8. Salicylic Acid (SA) and Direct Reproduction Assurance

SA treatments have been shown to enhance grain yield in crops facing heightened stress conditions [88,89]. However, to understand SA’s effects on reproduction under optimal growth conditions, we need to investigate the consequences of SA deficiency on plant growth and reproduction. These analyses have revealed that SA plays a significant role in suppressing shoot growth, photosynthesis, and seed yield [90]. SA also limits reproductive yield by combined senescence and flowering time acceleration [91,92].

The positive effects of SA on reproduction assurance are linked to its role in stress tolerance. The treatment of stressed plants with SA has been shown to enhance their ability to withstand stress, thereby ensuring a certain level of reproduction [88,89,93]. Notably, SA plays a crucial role in promoting tolerance to various abiotic stresses, including drought, salt, osmotic, and temperature stresses [88,89]. Additionally, SA is involved in plant defense against pathogens and contributes to immune responses [49]. It is worth mentioning that SA biosynthesis is significantly induced during pathogen infections and under various abiotic stress conditions [88,89,93].

## 5. Hormone Response Pathways and the Reproduction Trade-Off

Although hormone regulation encompasses several distinct processes, including biosynthesis, transport, and hormone activation and deactivation, it is the signaling pathways that ultimately dictate the strength and precision of hormone responses. Looking at it from this perspective, it is interesting to note that the CK signaling pathway operates differently than the signaling pathways of all other hormones. While the other hormone signaling pathways employ double-negative control mechanisms in which hormone perception leads to the inactivation of repressors of the hormone response, the CK signaling pathway involves a series of activation steps that incorporate a feedback-inhibition mechanism to regulate the intensity and duration of the response (Figure 5) [94,95,96,97,98,99,100,101,102,103]. The evolutionary “rationale” behind this difference in signaling pathway control mechanisms becomes clear when considering that the primary objective of CK is to maximize reproduction, which is the paramount trait favored by evolution. Achieving optimal reproductive success necessitates robust development and a signaling mechanism not constrained by excessive stimulation or overshooting concerns. Conversely, hormone modules that prioritize aspects other than reproduction maximization are likely to be more sensitive to environmental factors and actively suppressed under conditions of low environmental stress. The presence of an active repression mechanism ensures that the allocation of resources, from reproduction maximization to reproduction assurance, occurs only when necessary, effectively preventing any undesired deviation from the goal of reproduction maximization.

The emerging picture then reveals a scenario where CK action serves as the default mode, effectively driving the primary evolutionary process of reproduction maximization (Figure 6). Conversely, signaling pathways that facilitate reproduction assurance are actively suppressed when the plant’s well-being reaches an optimal or sufficient level. This intricate balance ensures that CK’s role in promoting reproduction maximization remains dominant unless elevated stress conditions compromise reproductive success. Notably, this dynamic interplay between reproduction maximization and reproduction assurance involves the direct suppression of CK action when plant survival is at stake. Hormones like AUXs, ABAs, GAs, and SLs, which are induced by stress and prioritize well-being, inhibit CK biosynthesis and signaling or enhance CK deactivation [17,18,104,105,106,107,108]. This regulatory mechanism ensures the precise allocation of resources, preventing any overshooting beyond the required threshold.

## 6. Origins and Evolution of Plant Hormone Modules

### 6.1. Origins

Although the exact origins of plant hormones remain unknown, one possible explanation for their emergence is linked to the synthesis of metabolites associated with beneficial physiological and developmental processes. If a particular metabolite consistently arises when a beneficial event like cell division occurs, it creates a stable environment for the evolutionary development of a mechanism capable of binding or detecting this metabolite. Subsequently, this mechanism can then evolve to stimulate further cell division, establishing yet another stable environment. Over time, this could lead to the evolution of a biosynthesis pathway that regulates the production of this metabolite, resulting in a cell division initiation or promotion mechanism that is responsive to the cell’s developmental and metabolic state. The structural similarities between CKs and nucleotides, the building blocks of nucleic acids, raise the possibility that CKs may have evolved from or interfaced with nucleotide-based signaling pathways and that CK is the original module for promoting reproduction [109]. Supporting this notion, the promotion of photosynthesis and reproduction by CK is observed even in microalgae, indicating its ancient nature [110].

Applying the same logical reasoning, it is worth noting that the primary natural auxin, indoleacetic acid (IAA), bears a structural resemblance to amino acids [111]. Interestingly, amino acids universally serve as signals for detecting the nutritional status of cells [112,113,114]. This observation leads to the tempting speculation that the auxin module, an essential well-being module, initially evolved to sense intracellular amino acid levels and respond to low levels by promoting nutrient uptake. Enhancing nutrient uptake in response to nutrient-deficient conditions is one of the primary functions of the current auxin module in higher plants [18]. Additionally, IAA, being an ancient molecule, has been demonstrated to promote cellular homeostasis in various primitive photosynthetic organisms [110,115]. Hence, it is plausible that, similar to CKs, auxins represent the original hormone module for promoting well-being, and the reciprocal interaction between CKs and auxins controlled the original balance between reproduction maximization and reproduction assurance.

Taking this perspective into account, it is essential to recognize that the absence of certain canonical hormone response pathways in ancestral algae does not necessarily negate the functional existence of the corresponding hormone modules in these plant ancestors [110]. While our current understanding of hormone response pathways is primarily based on well-studied plant species, it is plausible that ancestral algae possessed alternative mechanisms or pathways that facilitated hormone signaling and modulation of physiological processes related to growth, development, and reproduction. Indeed, it is plausible that hormone signaling in higher plants represents innovations in hormone modules that have emerged in response to environmental changes, including the transition from aquatic to terrestrial habitats during the emergence of land plants. The auxin response pathway is an illustrative example, as it has not been fully identified in algal ancestors [116]. However, it is worth noting that auxin does play a role in the growth regulation and physiology of algae. This suggests that while the specific mechanisms and components of the auxin response pathway may have evolved in higher plants, the fundamental influence of auxin on growth and development may have been conserved across plant lineages [116].

Interestingly, double-negative signaling, which involves the inhibition of inhibitory pathways, appears to be specific to hormonal modules of land plants. This unique characteristic may reflect the increased environmental stress that land plants experience compared to algae. In land plants, where resource allocation for well-being and reproduction assurance became more crucial, the pressure to optimize resource utilization became accordingly more pronounced. The demand for efficient resource allocation in the face of heightened stress likely drove the evolution of sophisticated mechanisms to ensure precise resource distribution without wastage. Double-negative signaling ensures that resources are allocated strategically and minimally to maintain plant well-being and reproduction, even under challenging environmental conditions.

Indeed, the emergence of new signaling innovations and new hormone modules in land plants can be attributed to their need to adapt to a more challenging environment characterized by drought, nutrient deficiency stresses, temperature fluctuations, increased UV-B exposure, and heightened oxidative stress, which is a combination of both increased number of ROS-generating stressors and the fact that, unlike algae, land plants cannot readily release reactive oxygen species (ROS) or ROS-generating compounds into their surroundings [23,117,118,119]. The transition from aquatic to terrestrial habitats brought about significant changes not only in abiotic stress factors but also in the biotic stress environment for land plants. Unlike algae, which primarily face challenges from waterborne pathogens and herbivores, land plants had to contend with a new array of biotic stresses, including diverse microbial pathogens, herbivorous insects, and grazing animals. The transition to land plants brought about another set of environmental challenges, which were indirectly caused by important plant developmental advancements, including the evolution of a vascular system that allowed for vertical plant growth. This advancement, for example, presented new growth obstacles related to shading, which in turn led to the development of a shade-avoidance response. To meet these new requirements, novel hormone modules likely emerged as existing modules were unable to adequately address conflicting demands. This highlights the crucial role of modularity in driving adaptation to changing environmental conditions [13].

### 6.2. Evolution

The hormone modules found in today’s higher plants are characterized by a high level of complexity, involving the intricate regulation of thousands of genes, as well as numerous post-transcriptional and post-translational controls. Understanding how these modules have maintained their functional focus on reproduction maximization or reproduction assurance, despite their increasing complexity and extensive interactions, may seem challenging. Nevertheless, suppose we view each hormone module as a tool with a central guiding principle. In that case, it becomes clear that the separation of functional focus needs to be maintained to accomplish the delicate balancing act of survival vs. reproduction through antagonistic or synergistic interactions. Under evolutionary pressures, mutations that compromise this focus would prove disadvantageous, whereas those that enhance it would be favored. This principle extends to mutations that improve modular interactions, leading to more refined and precise balancing acts. Hence, the advantageous approach of organizing life around a series of yin/yang balances between opposing motivations likely resulted in the preservation of the functional focus of distinct modules, even as their complexity increased. A case in point is the CK module that promotes photosynthesis and cell division in unicellular algae, representing an ancient version of photosynthetic life [110]. The main focal point of this CK module appears to have been preserved all the way to the highly complex multicellular angiosperms, wherein CK still promotes photosynthesis and reproduction.

## 7. Hormone Modules and Plant Domestication

Plant domestication inadvertently provides further proof of plant hormone modular function. The selection of high-yielding grain varieties has revealed that these varieties carry either loss-of-function in the GA module or gain-of-function in the CK module [72,120]. The role of CK in promoting reproduction is evident. However, cultivating plants that have an increased CK function necessitates the implementation of agricultural practices such as elevated fertilization and irrigation to mitigate drought and nutrient deficiency stress hypersensitivities associated with increased CK activity [18]. It is thus not surprising that the selection of CK gain-of-function for increased grain yields occurred in rice varieties grown under optimal hydration conditions (rice paddy agriculture) supplemented with high fertilization rates (e.g., night soil).

The GA loss-of-function success in improving grain yield is also evident, as GA is the sole well-being hormone that stimulates shoot growth and reduces the grain-to-vegetative biomass ratio [72]. Through the adoption of agricultural practices, like weed control, and the breeding of varieties with alterations in leaf shape and angle, the excessive shoot elongation required for the shade-avoidance response became less critical for overall shoot well-being [121]. On the other hand, GA’s action of destabilizing DELLA proteins, which are essential for plant stress tolerance, renders plants more susceptible to environmental stresses [122]. Consequently, partial loss of GA function emerged as a viable strategy for enhancing grain yields.

While these domestication events were inadvertent, future efforts are anticipated to involve deliberate modifications of hormone modules. For example, the targeted suppression of the CK module in roots leads to an enlarged root system, which could be favorable for root crops [123]. Promising results have also been observed by altering the BR module to improve grain yields [124]. As agricultural practices advance and become more adept at promoting overall crop well-being, there will be a decreasing necessity for specific well-being mechanisms that redirect resources away from reproduction or the growth of specific plant organs of agricultural importance. Consequently, these reproduction assurance modules could also serve as prospective targets for future crop improvements.

## Figures and Tables

**Figure 1 biology-12-01143-f001:**
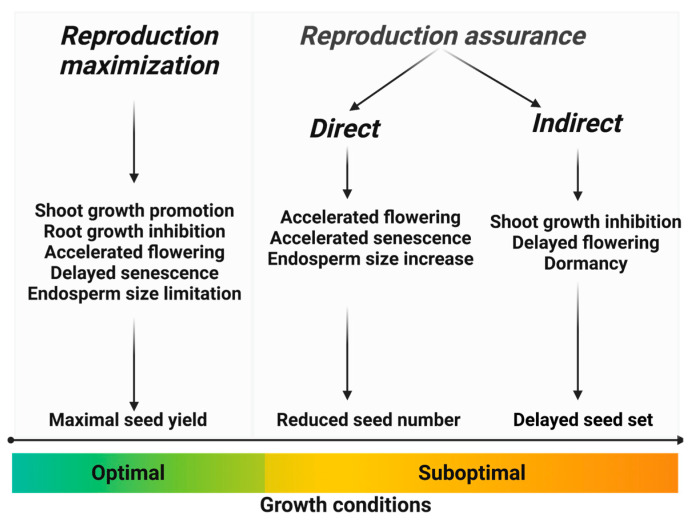
The impact of favorable and unfavorable growth conditions on the balance between reproduction maximization and survival (reproduction assurance), and the processes involved in maintaining this balance.

**Figure 2 biology-12-01143-f002:**
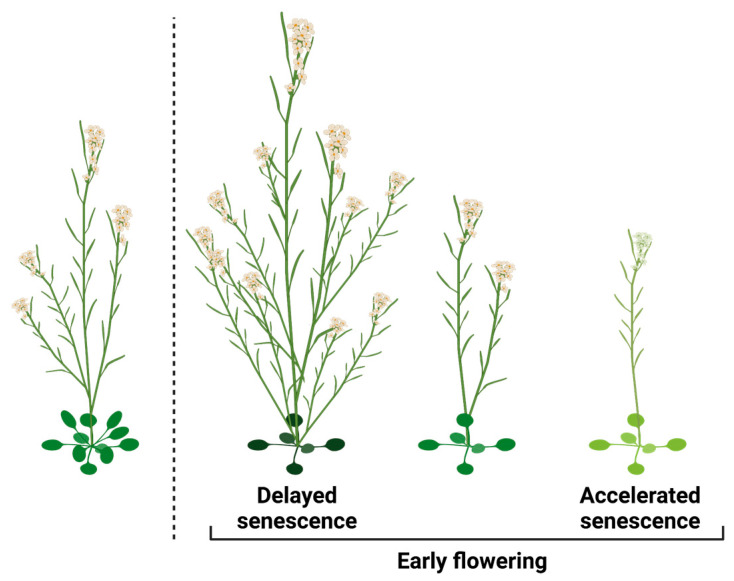
The effects of flowering time on reproduction. Early flowering can enhance reproduction, but its benefits are contingent on delayed senescence. On the other hand, certain forms of early flowering diminish overall reproduction, particularly when accompanied by accelerated senescence.

**Figure 3 biology-12-01143-f003:**
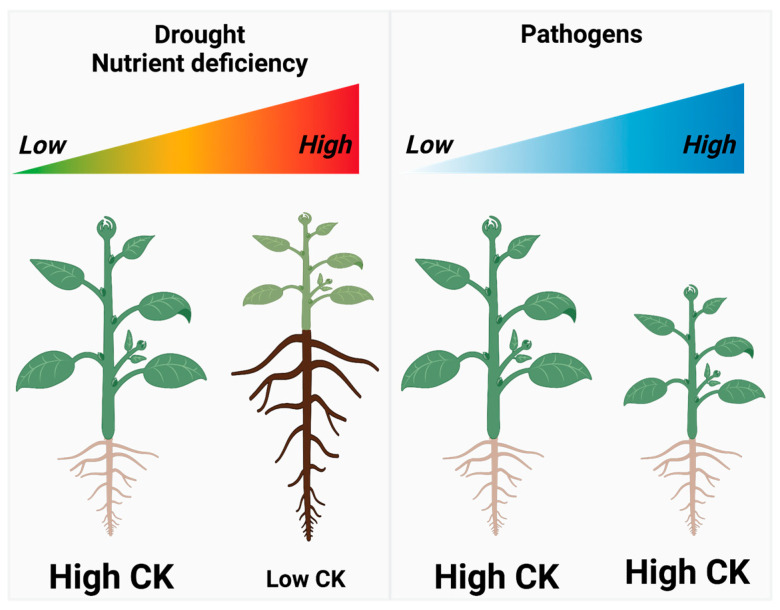
Examples of environmental stresses that define two categories of direct reproduction assurance by CK.

**Figure 4 biology-12-01143-f004:**
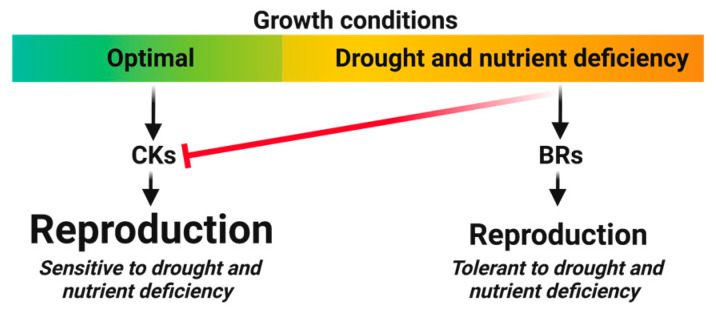
The schematic representation illustrates the relationship between cytokinins (CKs) and brassinosteroids (BRs) in environmentally controlled reproduction maximization. At optimal nutrient and hydration conditions, CK maximizes reproduction. CK action is down-regulated in response to drought or nutrient deficiency, while BR action is induced, resulting in continued reproduction maximization, albeit at a lower level.

**Figure 5 biology-12-01143-f005:**
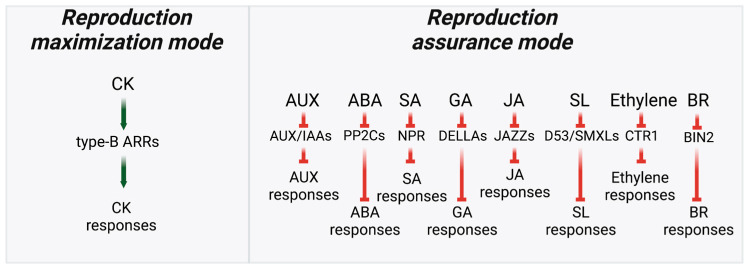
A simplified illustration of plant hormone signaling highlighting a distinction between direct activation in the CK pathway and double-negative-based activation in other hormone response pathways. CK signaling leads to the direct activation of response-activating type-B ARRs. Auxin (AUX), ABA, SA, GA, JA, SL, ethylene, and BR signaling leads to the inactivation of the respective response repressors: AUX/IAAs, PP2Cs, NPR, DELLAs, JAZZs, D53/SMXLs, CTR1, and BIN2.

**Figure 6 biology-12-01143-f006:**
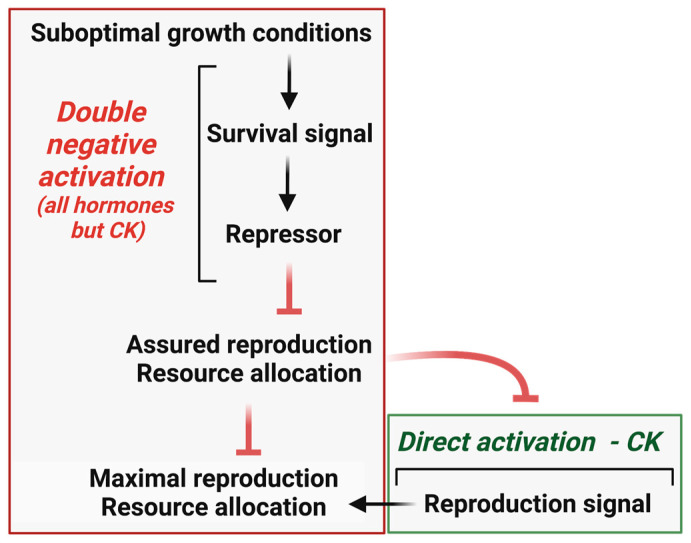
The schematic representation demonstrates the utilization of double-negative signaling and direct activation signaling to achieve maximal resource allocation for reproduction.

## Data Availability

Data sharing not applicable.

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
