# Peer review of "Plant Hormone Modularity and the Survival-Reproduction Trade-Off"

_biology, 2023, doi:10.3390/biology12081143_

Round 1

Reviewer 1 Report

General comments

This review addresses the functional role of plant hormones in the survival-reproduction trade-off. In contrast to mobile organisms that can escape from stress situations or non-plant organisms which undergo quiescence, for plants is crucial a correct balance between reproduction and survival. This is an original topic that was worth to describe and comment. For analyzing this subject, the authors carefully use the present literature introducing the interesting and novel concept of modularity. They describe the behavior of the different families of plant hormones and the role exerted by the different groups in the important switch between reproduction and survival in different stress conditions.

Overall, the review is well written and clear and provides an accurate explanation how plant hormones respond to different situations, In my opinion they fill a gap in the different role exerted by phytohormones in plants

Minor comments

1)     Even the paper is clear, sometimes concepts are repeated many times rendering the reading redundant, a more concise writing could facilitate the readers.

2)     The introduction of few more figures reducing the writing part could also facilitate the readers.

3)     Figure 2 should be introduced in the text.

Reviewer 2 Report

Title is too short elaborate more

Rewrite Abstract 

Introduction must be improved, add more reference 

Objective of the study is not clear rewrite again. 

Include table of the hormones generally involved and then screen specific from it. 

Reviewer 3 Report

This manuscript represents a full-length review on the subject of modularity of plant hormones and their role in the 8 trade-off between survival and reproduction. The manuscript includes all relevant recent publications, is well structured and gives the expected overview of our current understanding of this topic.

Reviewer 4 Report

The present article is intriguing and yet very problematic. Perhaps I do not share the broad biological background of the authors but, as a plant person, mainly trees, I have some serious objections. 

A) The Tradeoff. It's not survival vs. reproduction. It's the dilemma of resource allocation, whether to invest in vegetative growth or move towards reproduction ( see attached Abst.). Thus, I do not accept the distinction between 'maximizing' and 'assuring' reproduction, this is an unnecessary complication.                                                                                                   

B) The attempt to squeeze all other hormones into the 'assurance' framework is superfluous. For the sake of the present article it would be enough to demonstrate the role of CK and its unique regulatory function ( which should also be supported by by real experimental data ).                   

C) The discussion of the rest of the hormones is in certain cases rather sloppy, e.g. ethylene.                                                                                           

D) GA. More could be said about the role of GA in flowering, as related to stem elongation. GA promotes the bolting of              bi-annuals but inhibits flowering of fruit trees ( Acta Hort. 463: 201-8; 1998).

The survival of an individual tree does not depend upon sexual reproduction. Yet, the long term persistence of tree species requires an effective, asexual or sexual, means of reproduction. In the wild, most tree species reach reproductive maturity after a several decades of juvenility and even then, sexual reproduction appears sporadically, often in a mode of masting. Estimates of the reproductive allocation (= the percentage of annual photosynthate diverted towards sexual reproduction) in forest trees indicates a slow, gradual increase which may reach 50% in 'mast' years, but, on the average does not exceed 20%. The situation is different, however, in certain subtropical and tropical fruit trees (Citrus, Olive, Mango, Avocado), which invest a tremendous amount of resources in profuse flowering and fruiting. The reproductive allocation of a grapefruit tree has been evaluated as 79%. Some Citrus cultivars may collapse as a result of fruit overload and exhaustion of carbohydrate reserves. The rationale underlying this behavior might be that in their natural, original habitats these trees are exposed to environmental stresses, in particular drought, that threaten their survival. Thus, they divert all their resources towards sexual reproduction which is their highest priority. On the other hand, the survival of the aforementioned temperate and boreal forest trees is not endangered by environmental stresses; vegetative growth is their first priority and they maintain, on the average, a more moderate reproductive allocation.

Round 2

Reviewer 2 Report

The revised version is well organized and well revised. The authors embeded all the question. Consider the present form for publication. Thanks for considering me as reviewer.  

Author Response

Thank you

Reviewer 4 Report

My recommendation was 'major revision' but no revision has been done, just a cover letter and 2 new paragraphs. As said before, the MS is intriguing and should be published following a thorough, thoughtful revision.                            Let me repeat the major points : a) The awkward 'assurance' concept must be replaced by a better term. b) Revise the GA and ethylene sections ( see my earlier report) c) Support the presumed role of CK by real data.

Author Response

Reviewer 4:

My recommendation was 'major revision' but no revision has been done, just a cover letter and 2 new paragraphs. As said before, the MS is intriguing and should be published following a thorough, thoughtful revision. Let me repeat the major points:

Point 1: The awkward 'assurance' concept must be replaced by a better term.

Reply to Point 1: In our earlier response letter, we presented our viewpoint against employing the vegetative vs reproduction tradeoff terminology and advocated for embracing the "reproduction assurance" concept instead. The essence of this argumentation is replicated below for your consideration:

“We respectfully disagree with the reviewer's interpretation. The dilemma of resource allocation is described as "whether to invest in vegetative growth or move towards reproduction" implies that investing in vegetative growth is counterproductive to reproductive growth. On the contrary, vegetative growth can be considered a crucial stage necessary for reproduction maximization. This involves building sufficient photosynthetic capacity through shoot growth and establishing adequate nutrient and water uptake capacity through root growth. These attributes enable plants to sustain reproductive growth throughout their life cycle or achieve abundant reproduction in the later stages.”

Hence, our underlying assertion is that the primary objective throughout a plant's life cycle is reproduction. This premise is grounded in the logic that evolutionary forces favor species that emphasize reproduction. Accordingly, it becomes imperative to differentiate between strategies that prioritize the maximization of reproduction and those that make concessions to ensure reproduction at a reduced scale under challenging circumstances, necessitating a redirection of resources from reproduction to survival. Given the competing resource allocation demands of these strategies, it is logical to arrange them within a modular framework that permits interaction based on environmental requirements. This modular arrangement empowers plants to dynamically adjust to shifting conditions and optimize the allocation of resources as needed.

Could you please provide counterarguments that elucidate why this rationale is considered incorrect?

Point 2: Revise the GA and ethylene sections ( see my earlier report)

Reply to Point 2: In our previous response letter, we highlighted that a comprehensive review encompassing all aspects of hormone regulation extends beyond the scope of this article. Our initial response is reproduced below for your reference:

“The reviewer's observation is pertinent and emphasizes the importance of explicitly stating that the discussion of hormone functions was restricted. Indeed, throughout the manuscript, all hormone discussions were exclusively approached from the standpoint of their involvement in reproductive success. To address this oversight, we have included the following sentence in the first paragraph under Section "4. Hormones with Reduced Reproduction Trade-Off Ratios:

In this section, we present a succinct overview of hormone actions closely associated with reproduction. To delve deeper into the intricate mechanisms through which these hormones govern plant development and physiology, we direct readers to the numerous comprehensive reviews that are readily accessible. These reviews, due to their abundance, are not individually listed.”

Please share counterarguments elucidating the rationale behind considering this approach as incorrect.

Point 3: Support the presumed role of CK by real data.

Reply to Point 3: As highlighted in our previous response letter, our determination regarding the distinctive role of cytokinin stems from a comprehensive examination of the existing literature. Our assessment has been informed by a thorough analysis of numerous research papers and associated reviews, which provide pertinent citations to relevant research. The conclusions drawn in our study are firmly rooted in the empirical data presented within these research papers. We have reproduced our previous response to this matter for your convenience:

“The distinctiveness of cytokinin becomes apparent only when contrasted with other hormones. Thus, comparing these hormones and exploring their roles in plant reproduction was imperative. Differentiating between strategies that maximize reproduction and those that promote it on a more limited scale or with a delay proves valuable, as it establishes a framework for understanding hormone modularity. Throughout our manuscript, we have extensively cited various papers presenting experimental data supporting the conclusions drawn in this review. These references include both primary research papers and review articles that cite relevant research.”

Could you please offer counterarguments explaining the rationale behind not considering the data presented in these referenced research papers as "real data"?

Round 3

Reviewer 4 Report

Even if authors insist on staying with the 'assurance' terminology, they should have addressed the minor points in my 2nd Review - from my memory - * Revise the GA/flowering section * Provide a comprehensive (referenced!) paragraph regarding ethylene (fruit ripening etc.) * Present real quantitative data demonstrating the presumed CK effects, not just conceptual models. Since the Authors decided to discuss all plant hormones, accuracy is required. This revision must be completed prior to acceptance of the MS for publication in 'Biology'.